# AAA+ Molecular Chaperone ClpB in *Leptospira interrogans*: Its Role and Significance in Leptospiral Virulence and Pathogenesis of Leptospirosis

**DOI:** 10.3390/ijms21186645

**Published:** 2020-09-11

**Authors:** Sabina Kędzierska-Mieszkowska, Zbigniew Arent

**Affiliations:** 1Department of General and Medical Biochemistry, Faculty of Biology, University of Gdańsk, 80-308 Gdańsk, Poland; 2University Centre of Veterinary Medicine, University of Agriculture in Krakow, 30-059 Krakow, Poland; zbigniew.arent@urk.edu.pl

**Keywords:** ClpB, *Leptospira*, leptospirosis, molecular chaperone, pathogen–host interactions, virulence factors

## Abstract

Bacterial ClpB is an ATP-dependent disaggregase that belongs to the Hsp100/Clp subfamily of the AAA+ ATPases and cooperates with the DnaK chaperone system in the reactivation of aggregated proteins, as well as promotes bacterial survival under adverse environmental conditions, including thermal and oxidative stresses. In addition, extensive evidence indicates that ClpB supports the virulence of numerous bacteria, including pathogenic spirochaete *Leptospira interrogans* responsible for leptospirosis in animals and humans. However, the specific function of ClpB in leptospiral virulence still remains to be fully elucidated. Interestingly, ClpB was predicted as one of the *L. interrogans* hub proteins interacting with human proteins, and pathogen–host protein interactions are fundamental for successful invasion of the host immune system by bacteria. The aim of this review is to discuss the most important aspects of ClpB’s function in *L. interrogans*, including contribution of ClpB to leptospiral virulence and pathogenesis of leptospirosis, a zoonotic disease with a significant impact on public health worldwide.

## 1. Introduction

*Leptospira interrogans* is one of the many pathogenic species of the *Leptospira* genus that can cause a serious disease in mammals known as leptospirosis. This disease belongs to the bacterial zoonoses, i.e., infections that can be transmitted from animals to humans. Leptospirosis has a significant impact on public health worldwide. More than one million human cases of severe leptospirosis and approximately 60,000 deaths from this disease have been estimated to occur annually [1]. Leptospirosis also generates huge economic losses in a number of countries due to reproductive disorders in farm animals (e.g., cattle, sheep, pigs, and horses) [2,3,4,5]. However, until now, molecular mechanisms of both, the leptospiral virulence and pathogenesis of leptospirosis remain largely unknown.

Similar to other pathogenic bacteria, *L. interrogans* has developed numerous strategies to interact with its mammalian hosts and to rapidly adapt to new environmental conditions, including oxidative stress, which is associated with the host immune response. These strategies allow this pathogen to subjugate the host’s cellular factors and functions, and hence effectively establish itself within the host. To curb infection, the attacked host activates its defense and protective mechanisms, i.e., the innate immune response, involving neutrophils and macrophages, and adaptive response involving B and T cells [6]. Thus, interactions of bacterial pathogens with the host’s cells lead to responses in both the pathogen and its host. Virulence factors produced by pathogenic bacteria, including *L. interrogans*, play an important role in conquering the host cells and subverting the host’s cellular processes to the pathogen’s advantage [7,8]. Thanks to these factors, bacterial pathogens to interfere with the host’s physiological responses, and thus are able to succeed in the host–pathogen battle and cause the disease. Therefore, identification and characterization of bacterial virulence factors, and understanding their role in the pathogen–host protein interactions are fundamental for revealing the molecular mechanisms of infectious diseases, such as leptospirosis, and also for developing effective strategies to combat bacterial infections. To date, by using animal infection models and in vitro macrophage models very few proteins have been identified as potential virulence factors of *L. interrogans* (Table 1), including the molecular chaperone ClpB which belongs to the Hsp100/Clp subfamily of the AAA+ ATPases (*ATPases associated with diverse cellular activities*) [9]. It is important to note that the Hsp100 members are found in bacteria, protozoa, yeast, and plants, but not in animals and humans. This review focuses on the role of ClpB in pathogenic *L. interrogans* and its importance during leptospiral infections. We suggest that ClpB is one of the *L. interrogans* factors which are necessary to overcome the oxidative stress produced by the host defense mechanisms and is needed to invade the host and cause disease. Of note, since ClpB does not exist in animal and human cells, it seems to be a promising drug target for anti-leptospiral strategies, which is particularly important in the current era of increasing antibiotic resistance among numerous pathogenic bacteria.

## 2. *Leptospira*, Leptospirosis, and Cross-Talk between Pathogenic *Leptospira* and Their Host

Leptospires belong to spirochaetes and initially, based on 16 S rRNA gene relationships, 21 distinct species were identified [21]. However, employing whole-genome sequencing (WGS) as a genus-wide strain classification and genotyping tool has recently led to reclassifying the *Leptospira* genus into a total of 64 *Leptospira* species. Two major clades and four subclades were identified, i.e., clades P and S, and subclades P1—formerly described as the pathogen group, P2—formerly intermediate group, S1—saprophyte group, and S2—a new saprophyte subclade [22]. Saprophytic species of *Leptospira* exist in an environment associated with soil or surface waters, and generally they do not cause disease in animals and humans [23]. Intermediate species of *Leptospira* exhibit moderate pathogenicity in both animals and humans. Pathogenic *Leptospira* spp., including *L. interrogans*, can efficiently infect many animals and can also be transmitted to humans. Sources of pathogenic species mostly include wild (rodents) or domestic animals (dogs, cattle, pigs, sheep, horses), which harbor these spirochaetes in the proximal renal tubules of the kidneys and chronically excrete leptospires into the environment via urine [24]. Thus, the transmission of leptospirosis to humans primarily occurs through direct contact with the urine of infected animals or indirectly through contact with a urine-contaminated environment, i.e., water or moist soil. In humans, the disease varies from an asymptomatic flu-like illness to an acute life-threatening infection (known as Weil’s disease), with pulmonary hemorrhage, myocarditis, and kidney and liver failure [24]. In animals, leptospirosis may not demonstrate any clinical symptoms. However, this disease may indirectly add to other problems, such as decreased milk production in cattle, reproductive disorders due to abortions, and stillbirths in cattle, sheep, and pigs, or a persistent eye condition in horses, i.e., periodic ophthalmia or moon blindness [2,4,5].

The preferred portals of entry for pathogenic *Leptospira* spp. are either interruptions in the integrity of the skin, including cuts and scrapes, or mucous membranes, such as the conjunctival, oral, and genital areas [25]. Shortly after penetration of the host’s tissue barriers, the pathogen enters the bloodstream where it persists during the leptospiremic phase of the disease [25]. Once this pathogen accesses the bloodstream, it multiplies and disseminates to some organs, e.g., the liver is the major target organ in human leptospirosis [25]. Pathogenic leptospires are highly invasive and cunning bacteria; they can evade the host immune response during the initial phase of infection and succeed in infection. Unfortunately, the evasion mechanisms used by the pathogenic *Leptospira* spp. are still unclear. It is known that these spirochaetes are resistant to the host’s complement system, which is its first line of defense against pathogenic bacteria, and therefore plays a key role in the host–pathogen battle [26]. It has been demonstrated that the *L. interrogans* LenA and LenB surface-exposed proteins can bind to the human complement regulators, including factor H, thereby contributing to this pathogen’s resistance to the complement-mediated killing during leptospiremic phases of the disease [27,28,29]. LcpA is another outer membrane protein of pathogenic *Leptospira* spp. that is able to bind some of the human complement regulators, including vitronectin [30,31]. Further, pathogenic *Leptospira* spp. can bind human plasminogen (PLG) that is a key component of the host fibrinolytic system [29,32,33]. Acquisition of plasminogen (PLG)/plasmin in order to destroy the host’s complement proteins is another evasion strategy used by pathogenic *Leptospira*. It has been demonstrated that members of the family of leptospiral immunoglobulin-like (Lig) proteins (i.e., LigA and LigB), which are found only in the pathogenic *Leptospira* spp., can interact with PLG, leading to its activation and destruction of the host complement components [34]. In addition, it is possible that leptospires use their own proteases for inactivation of the host’s complement system components [29]. Thus, pathogenic leptospires have evolved appropriate factors to take control of the complement activation and to efficiently evade the host’s innate immunity.

Adhesion of the pathogen to the host’s extracellular matrix (ECM) proteins and to the host cells is also a crucial step in leptospiral infections. This step is decisive for successful colonization and invasion of the host tissues. For this purpose, pathogenic leptospires can use, for example, a set of six Len proteins: LenA–LenF [28], which exhibit affinities for laminin and fibronectin. Furthermore, the above-mentioned LigA and LigB proteins may be involved in interactions with the host ECM proteins and in leptospiral adhesion to the host cells [35,36]. Interestingly, it has been demonstrated that binding of recombinant LigB to fibrinogen inhibits fibrin formation, which could facilitate pathogenic *Leptospira*’s entry into the bloodstream, dissemination in the host tissues, and thus further infection by impairing the healing process [36]. Besides the above-mentioned ECM- and PLG-binding proteins, other *Leptospira* proteins have a significant impact on the infection’s progress, including virulence factors described and listed in Table 1. These factors increase *Leptospira*’s chance of invasion and survival within the host. The mobility of *Leptospira* provided by endoflagellum (which is unique among bacteria), certainly plays an important role in its penetration of the host tissues and its dissemination. The lipopolysaccharide (LPS) of pathogenic *Leptospira* spp., an important virulence determinant (Table 1), is important primarily because of its signaling properties. It has been demonstrated that leptospiral LPS activates the plasma membrane Toll-like receptor 2 (TLR2) instead of TLR4 for signaling in humans cells, whereas in mice, which are resistant to acute leptospirosis, both TLR2 and TLR4 recognize leptospiral LPS [37]. This is due to a difference in recognition of the lipid A part of the leptospiral LPS by these TLRs—only murine TLR4 is able to recognize the lipid A moiety [37]. Further, the results of Chassin and colleagues [38], obtained for mice, have demonstrated that TLR4 is required for B cell activation and production of specific IgM directed against pathogenic *Leptospira*. Therefore, it is responsible for their early clearing from the bloodstream. Thus, the fact that the human TLR4 does not significantly participate in leptospiral recognition could indicate a bacterial mechanism of avoiding recognition by the human immune system [37,39]. Interestingly, according to the recent in vitro study of Bonhomme et al. [40], although leptospiral LPS is recognized by the murine TLR4, it avoids the TRIF adaptor arm of the TLR4 response. This may prove that pathogenic leptospires have evolved yet another immune evasion mechanism linked to their LPS. Further, Bonhomme and colleagues [40] suggest that the limited LPS recognition in mice could be the key to the chronicity of leptospires in rodents. In addition, LPS enables leptospires to adapt to temperature changes [41]. Apart from these data, there is increasing evidence that the ClpB molecular chaperone contributes to leptospiral infections. Therefore, the next sections of this review are dedicated to this chaperone and its role in pathogenic *L. interrogans.*

In summary, pathogenic *Leptospira* spp. have evolved an arsenal of different and sometimes multifunctional molecules to invade and colonize the host tissues and to protect them against the host’s physiological responses.

## 3. ClpB Disaggregase—Its Structure and Mechanism of ClpB-Mediated Protein Disaggregation and Reactivation

During the past decades, many studies have focused on the reactivation of the stress-aggregated proteins by a bi-chaperone system consisting of the major chaperone system DnaK/DnaJ/GrpE and ClpB [42,43,44,45,46,47,48]. It has been demonstrated that ClpB is crucial for bacterial survival under environmental stresses, including thermal and oxidative stresses [48,49,50]. Similar to other Hsp100 proteins, ClpB forms ring-shaped hexamers in the presence of ATP [45,51,52,53], with a narrow channel (pore) at the center of the hexameric ring. Each ClpB protomer is composed of four domains (Figure 1): an N-terminal domain responsible for binding and recognition of protein substrates, two AAA+ ATP-binding modules (nucleotide-binding domain 1 (NBD-1), nucleotide-binding domain 2 (NBD-2)), and a unique coiled-coil middle domain (MD) inserted at the end of NBD-1 and involved in functional interactions with DnaK that are required for efficient protein disaggregation [54,55]. According to the recent findings [56], the mechanism of protein disaggregation and reactivation mediated by ClpB couples the ATP hydrolysis with processive extrusion of looped polypeptides and their translocation through the central channel of ClpB hexamer. In this model of ClpB action, polypeptide loop extrusion is proposed as the mechanistic basis of protein disaggregation.

To date, ClpB chaperone from the well-characterized *Escherichia coli* and its yeast homolog, the Hsp104 protein, have been the most intensely explored Hsp100 proteins because of their unique protein disaggregation activity. A number of studies have aimed to understand how the ClpB/Hsp104 molecular machine functions to reverse protein aggregation as this machinery provides hope for the development of new and more effective therapeutic strategies to combat both the neurodegenerative and systemic diseases related to protein aggregation. However, as discussed below, this is not the only direction of research associated with ClpB.

## 4. The Role of ClpB in Pathogenic *L. interrogans*—Another Side of This Chaperone Function

Numerous studies have revealed that ClpB is also involved in supporting the virulence of some bacterial pathogens, including the pathogenic spirochaete *L. interrogans* [9,57,58,59,60], but However, the specific function of ClpB in bacteria during infection of their hosts is still unexplored. To date, it has been demonstrated that the loss of ClpB function in *L. interrogans* cells results in bacterial growth defects under stress conditions, such as nutrient limitation, and heat and oxidative stresses [9]. Importantly, ClpB deficiency made this pathogen avirulent in an animal model of acute leptospirosis (in gerbils) as compared to its parental strain [9]. Thus, it has been shown that ClpB is essential for leptospiral survival under various stress conditions and also during infection of mammalian hosts. The role of stress and stress response in the host–pathogen interactions, and also the participation of ClpB in the pathogen’s stress response, designate a new field of research on molecular mechanisms of leptospirosis. So far, we have demonstrated that ClpB is not only produced during leptospiral infection but is also able to elicit immune responses in the infected animals, further confirming its involvement in the pathogenicity of *L. interrogans* [61]. At this point, it is worth mentioning that virulence factors are often immunogenic and responsible for the acquired immunity that protects against a given disease [62]. Moreover, it is known that despite their cytosolic localization, some molecular chaperones, including ClpBs from pathogenic bacteria (e.g., *Mycoplasma pneumoniae* and *Francisella tularensis*), are strongly immunogenic in a number of bacterial infections [57,63,64]. Therefore, exposure of bacterial heat-shock proteins (Hsps) to the host’s immune system is highly possible during infection.

Further, we have shown [65] that the *L. interrogans* ClpB can assemble into hexamers in a nucleotide-dependent manner, similar to other well-characterized bacterial ClpBs [52,53,66,67], and exhibits an aggregate-reactivation activity that may contribute to the survival of *L. interrogans* under the host-induced proteotoxic stress (e.g., high body temperature or oxidative stress) that is likely to induce aggregation of the pathogen’s proteins [68,69]. To succeed in an infection, the pathogen has to successfully respond to the host-induced stress because only then it can survive and multiply in the host cells, leading to the development of the disease; thus, ClpB’s aggregate-reactivation activity may ensure the pathogen’s stress survival. It is interesting that *L. interrogans* ClpB may mediate protein disaggregation independent of assistance from the DnaK chaperone system [65]. Involvement of ClpB in the stress response of *L. interrogans* demonstrated in [9] and the above-mentioned results of our studies suggest that this chaperone may be one of the key mediators of stress resistance (a prerequisite for disease pathogenesis), because of its aggregate-reactivation activity [65].

To further elucidate ClpB’s role in leptospiral virulence and pathogenesis of leptospirosis, we have identified putative substrates for this chaperone, i.e., proteins that may require the assistance of ClpB in *L. interrogans* cells under stress conditions, including host-induced stress [70]. Our proteomic study and the use of a “substrate trap” variant of ClpB with mutations within the Walker B motif of both ATP-binding domains (E281A/E683A), which is deficient in ATP hydrolysis and forms stable complexes with its protein substrates, have revealed that the majority of ClpB-interacting proteins (~41% out of 68 identified proteins) are associated with key metabolic pathways, such as the TCA cycle (Tricarboxylic acid cycle), glycolysis–gluconeogenesis, or amino acid and fatty acid metabolism [70] (Figure 2). Metabolic enzymes certainly have a significant impact on the growth of *L. interrogans* and its virulence. It has been demonstrated that some leptospiral genes involved in metabolic processes were up-regulated upon exposure of *L. interrogans* to blood serum [71]. The other identified proteins were associated with ribosome biogenesis, translation, redox homeostasis, proteolysis, or cell wall and membrane biogenesis. Based on these results, we postulate that ClpB supports the virulence of *L. interrogans* mainly by protecting the conformational integrity and catalytic activity of multiple metabolic enzymes, which are sensitive to stress and prone to aggregation. Hence, ClpB is responsible for maintaining energy homeostasis in pathogenic *L. interrogans*. Due to its contribution to oxidative stress survival, ClpB is among the *Leptospira* factors that may ensure this pathogen’s escape from the host’s defense mechanisms and help it to avoid killing by phagocytes (mainly macrophages).

When discussing the role of ClpB in pathogenic *L. interrogans*, it is worth recalling a recently published work by Kumar et al. [72], where the combined approach of in silico algorithms, network theory, and functional annotations was applied to analyze and understand interactions between the *L. interrogans* and human proteins. This analysis revealed that ClpB is one of the *Leptospira* hub proteins interacting with numerous human proteins (e.g., collagen type IV alpha 1 chain, thioredoxin-interacting protein, translational elongation factor 1 alpha 1, plasminogen precursor, or adenylyl cyclase-associated protein 1). Figure 3 presents, in our opinion, the most important potential ClpB–human protein interactions and their possible role during leptospiral infections. Of note, another AAA+ ATPase and chaperone—ClpY (HslU) that forms a protease complex with ClpQ (Hslv) and is also involved in the leptospiral survival and virulence [69], interacts with only one human protein, i.e., inhibitor of growth protein 5 (ING5) that belongs to the ING protein family and functions as a type-II tumor suppressor gene. It can be noted that among human ClpB interactors, there are components of extracellular matrix (ECM) and the host plasma that could indicate ClpB’s participation in adhesion of *Leptospira* to the host cells and in plasminogen acquisition. A definite answer to these possibilities requires further research. Furthermore, in the same study, ClpB was found among seven leptospiral proteins involved in inactivation of T cells, and the adaptive and innate immunity and inflammation, thereby resulting in damage to the defense mechanisms of the host. Thus, Kumar and colleagues’ prediction [72] may be the bracing clamp in the current research on the role of ClpB in *L. interrogans* that points out that ClpB may participate in the immune evasion mechanisms developed by this pathogen. Further studies are needed to fully explore and understand the role of ClpB in leptospiral virulence and pathogenesis of leptospirosis, a disease that is a common problem for humans and animals throughout the world.

## 5. Conclusions

The aim of this review was to underline the potential importance of the molecular chaperone ClpB/Hsp100 in infections caused by the pathogenic spirochaete *L. interrogans*. Evidence accumulated so far indicates that ClpB’s unique protein disaggregation activity may play an important role in this pathogen’s adaptation to mammalian hosts. This chaperone is among the factors that may ensure stress survival of the pathogenic *Leptospira* species, especially during oxidative stress associated with the host immune responses. By contributing to the pathogens’ stress survival, ClpB is involved in a molecular dialogue (cross-talk) between the pathogen and its host during the infection process. A better understanding of mechanisms by which pathogenic *Leptospira* spp. respond to the host-induced stress may be useful for developing new strategies to combat leptospiral infections, and since ClpB does not exist in animal and human cells, it might become an attractive target in designing new anti-leptospiral agents that could be more effective than the currently used antibiotics. It should be mentioned that Hsps, whose function is supported by energy from ATP hydrolysis seem to be promising druggable targets [73,74,75]. For this reason small-molecule inhibitors of the ATPase activity of Hsps that mimic ATP may be evaluated for their efficacy in clinical trials. So far, the Hsp70 and Hsp90 ATPases have been presented as the most promising drug targets in human diseases, including infectious diseases caused by protozoan parasites [73,74,75]. There is a high probability that the ClpB ATPase will also show great potential as a druggable target.

## Figures and Tables

**Figure 1 ijms-21-06645-f001:**
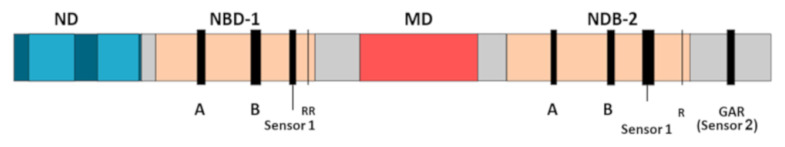
Domain organization of the ClpB monomer [based on 61]. The monomer is composed of the following domains: N-terminal domain (ND), nucleotide-binding domain 1 (NBD-1), middle domain (MD), and nucleotide-binding domain 2 (NBD-2). All characteristic and conserved motifs of the AAA+ ATPases coordinating ATP binding and hydrolysis, such as the Walker A: GX_4_GKT/S (A), Walker B: Hy_2_DE (B), sensor 1, sensor 2 (GAR), and the arginine fingers (R), are also indicated.

**Figure 2 ijms-21-06645-f002:**
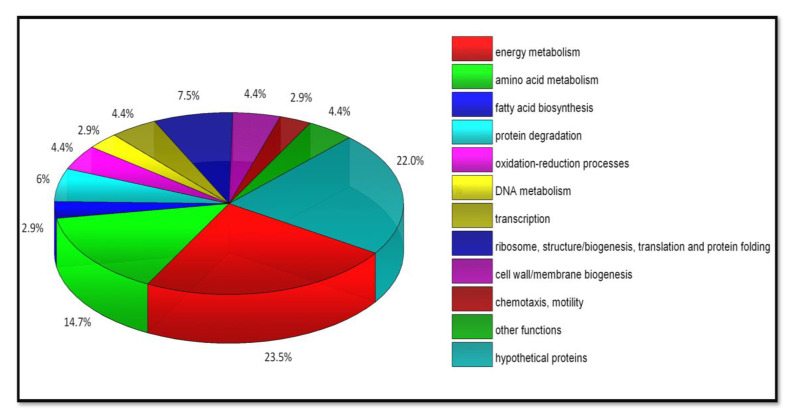
Functional classification of putative protein substrates of *L. interrogans* ClpB. The pie chart was created using the OriginLab software (OriginPro2016, Northampton, MA, USA) and shows the percentage distribution of the identified ClpB-interacting proteins among different biological processes [70].

**Figure 3 ijms-21-06645-f003:**
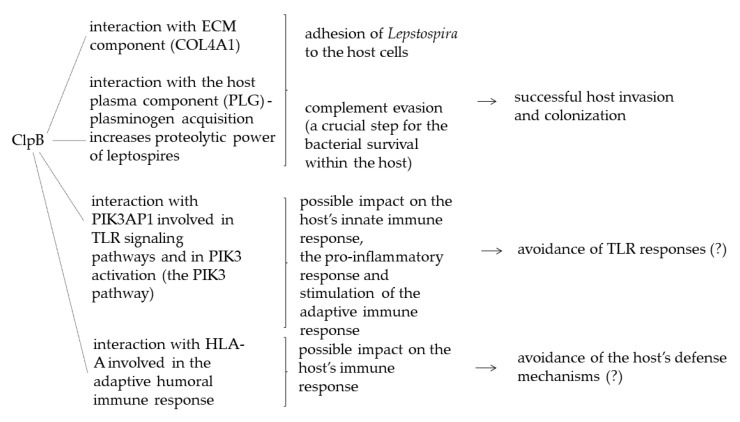
The most important potential ClpB–human proteins interactions and their possible role during leptospiral infections [72].

**Table 1 ijms-21-06645-t001:** Known virulence factors of *L. interrogans* identified in animal models of acute leptospirosis and in in vitro macrophage models * [6,9,10,11,12,13,14,15,16,17,18,19,20].

Factor Name	Description/Function during Infection
LPS (lipopolysaccharides)	Cell wall component; a key virulence factor that is involved in this pathogen’s adaptation to temperature changes
Adenylate/guanylate cyclase (AGC)	Its activity elevates intracellular cAMP in macrophages, therefore AGC may reduce the host innate TNF response
ClpB	Molecular chaperone Hsp100; probably contributes to adaptive response of pathogenic *Leptospira* spp. to the host-induced stress; possible participation in phagosomal escape and dissemination of this pathogen in the host tissues
FlaA2, FliY, FliM, FcpA	Flagellar components responsible for pathogen motility that is essential for its entry and dissemination in host tissues
HtpG	Molecular chaperone Hsp90 essential for virulence in the hamster model of acute leptospirosis; its function during infection is unknown; due to its chaperone activity, it may contribute to adaptive response of pathogenic *Leptospira* spp. to the host-induced stress
HemO (heme oxygenase)	Heme-degrading enzyme required for iron acquisition; iron is essential for successful leptospiral colonization of mammalian hosts
KatE	Catalase, located in the periplasmic space of *Leptospira* spp. cell wall that contributes to ROS resistance; it is possible that KatE participates in escape of the pathogen from neutrophils and macrophages by detoxification of the ROS produced by these phagocytes
LB139 (sensor protein)	Regulates gene expression, including genes encoding proteins required for motility and chemotaxis
LoA22 (OmpA-like protein)	Outer membrane protein of unknown function; it plays an important role in the acute model of infection
LruA	Lipoprotein mainly located in the inner membrane of the leptospiral cell; it interacts with ApoA-I, a component of the host innate immune system
Mce (mammalian cell entry protein), ColA (collagenase A)	Outer membrane proteins which are most likely involved in invasion and transmission of *Leptospira* spp.
Phospholipase C * (PI-PLC) LB361 gene product)	Hydrolyzes phosphatidylinositol-4,5-bis phosphate; it contributes to increased intracellular level of free calcium ions, causing death of infected macrophages

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
