# Peer review of "AAA+ Molecular Chaperone ClpB in *Leptospira interrogans*: Its Role and Significance in Leptospiral Virulence and Pathogenesis of Leptospirosis"

_ijms, 2020, doi:10.3390/ijms21186645_

Round 1
Reviewer 1 Report
The topic is relevant and interesting. In this review, the authors have tried to discuss recent development and key issues related to roles and significance of AAA+ and molecular chaperone ClpB in virulence and pathogenesis of leptospirosis. However, the review on solely of ClpB and lacks a relative discussion on how this protein has a potential role than other members of the family. Therefore, the authors need to share elaborative views to male this article more informative. Though the researchers made commendable work on molecular chaperones, more reviews in the field is well intentional as member candidates of the family - molecular chaperones play an indispensable role for the cell survival.
The introductory part may describe portal of entry, mode of transmission, host-pathogen interactive mechanism and clinical manifestation of leptospirosis.
Authors may add information indicating the functions of chaperone systems in general than specific, based on well studied other chaperone proteins such as heat shock protein (HSP), DnaJ, DnaK and GroEL and role of these with specific reference to innate immunity, induction of both humoral and cell-mediated, survival of cell during adverse conditions, i.e. drought, salinity, chemicals, cold and hot, and survival mechanism of cell through formation of cell aggregates wither self or combination with other microbes thus would enhance scope of this review (L159-162).
Authors hypothesize and conclude that ClpB could become a drug target since it is absent in the host cells but there is no discussion on how it compared with other chaperones as potential drug targets (L285-289).
Although the authors have discussed bacterial survival under adverse environmental stresses but they did not mention about biofilm formation process in pathogenic leptospira. Although English language is not first language for authors but expression and grammatical part should be improved.
Author Response
Response to Reviewer 1 Comments
Point 1. The topic is relevant and interesting. In this review, the authors have tried to discuss recent development and key issues related to roles and significance of AAA+ and molecular chaperone ClpB in virulence and pathogenesis of leptospirosis. However, the review on solely of ClpB and lacks a relative discussion on how this protein has a potential role than other members of the family. Therefore, the authors need to share elaborative views to male this article more informative. Though the researchers made commendable work on molecular chaperones, more reviews in the field is well intentional as member candidates of the family - molecular chaperones play an indispensable role for the cell survival.
Response 1: We would like to thank the Reviewer for all comments and suggestions.
Leptospires, especially pathogenic species, are still a major challenge for researchers. Their biology still hides many secrets. The role of molecular chaperones in leptospiral survival, virulence, and pathogenesis of leptospirosis still needs to be explored. In this review, we have focused on the role of ClpB because extensive evidence indicates that ClpB supports virulence of pathogenic spirochaete Leptospira interrogans (lanes 201-212). To our knowledge the role of other major chaperone such as DnaK, DnaJ, or GroEL has not been demonstrated yet in leptospiral virulence or in pathogenesis of leptospirosis. Therefore, other major chaperones are not included in this manuscript. Besides, in our opinion, their role goes beyond the scope of our work. We would like to emphasize once again that the work is dedicated exclusively to the ClpB molecular chaperone from L. interrogans. We hope that all data for this chaperone have been taken into account.
Point 2. The introductory part may describe portal of entry, mode of transmission, host-pathogen interactive mechanism and clinical manifestation of leptospirosis.
Response 2: We would like to note that the above mentioned issues have been already described in the manuscript (please see Section 2: Leptospira, leptospirosis and cross-talk between pathogenic Leptospira and their host).
Point 3. Authors may add information indicating the functions of chaperone systems in general than specific, based on well studied other chaperone proteins such as heat shock protein (HSP), DnaJ, DnaK and GroEL and role of these with specific reference to innate immunity, induction of both humoral and cell mediated, survival of cell during adverse conditions, i.e. drought, salinity, chemicals, cold and hot, and survival mechanism of cell through formation of cell aggregates wither self or combination with other microbes thus would enhance scope of this review (L159-162).
Response 3: As mentioned above, the role of DnaJ, DnaK, and GroEL in Leptospira spp. still needs to be explored. For this reason these chaperones are not included in this manuscript.
Point 4. Authors hypothesize and conclude that ClpB could become a drug target since it is absent in the host cells but there is no discussion on how it compared with other chaperones as potential drug targets (L285-289).
Response 4: It should be mentioned that Hsps whose function is supported by energy from ATP hydrolysis seem to be promising druggable targets [73-75]. For this reason small-molecule inhibitors of the ATPase activity of Hsps that mimic ATP may be evaluated for their efficacy in clinical trials. So far, the Hsp70 and Hsp90 ATPases have been presented as the most promising drug targets in human diseases, including infectious diseases caused by protozoan parasites [73-75]. There is a high probability that the ClpB ATPase will also show great potential as a druggable target.
Point 5. Although the authors have discussed bacterial survival under adverse environmental stresses but they did not mention about biofilm formation process in pathogenic leptospira.
Response 5: Indeed, some pathogenic Leptospira spp. are able to produce a biofilm. However, a recent report indicates that biofilm production does not seem to promote leptospiral virulence and does not lead to more effective colonization of the kidneys (Thibeaux, R., Soupé-Gilbert, M., Kainiu, M. et al. The zoonotic pathogen Leptospira interrogans mitigates nvironmental stress through cyclic-di-GMP-controlled biofilm production. npj Biofilms Microbiomes 6, 24 (2020). https://doi.org/10.1038/s41522-020-0134-1). However, biofilm-forming leptospires were less sensitive to extracellular stresses. It is important to mention that in the case of saprophytic Leptospira species, biofilm production is correlated with the ability to persist in the environment (Thibeaux et al., 2020). This fact may explain the environmental persistence of pathogenic L. interrogans.
Furthermore, the role of ClpB in biofilm formation has yet to be investigated, therefore this issue is not described in our manuscript.
Point 6. Although English language is not first language for authors but expression and grammatical part should be improved.
Response 6: We have tried our best to improve the English language. Therefore, we have proofread the manuscript again, but because the reviewer gave no examples, we are unclear about what changes are requested.
Reviewer 2 Report
Dear author
I have no comments to add, except I miss one graphic or drawing with the Clp interactions where it could be possible, in which patways is involved and, how could affect to, e.g. host immune system
Nothing else
Author Response
Response to Reviewer 2 comments
Point 1. I have no comments to add, except I miss one graphic or drawing with the Clp interactions where it could be possible, in which pathways is involved and, how could affect to, e.g. host immune system.
Nothing else
Response 1: We would like to thank the Reviewer for the positive review.
As suggested by the Reviewer, in the revised manuscript, we have added Figure 3 (lane 277) which presents, in our opinion, the most important potential ClpB-human proteins interactions and their possible role during leptospiral infections, i.e. their impact on the host’s immune response, and the following additional information in the main text is also included (lines: 261-262): Fig. 3 presents, in our opinion, the most important potential ClpB-human proteins interactions and their possible role during leptospiral infections.
Reviewer 3 Report
I recommend this manuscript for publication in IJMS, although before proceeding, I encourage authors to revise some minor issues listed below.
- Page 1, line 32: Please replace “60 000” to “60,000”.
- Page 6, line 219: Please replace “nucleotide- dependent” to “nucleotide-dependent”.
Author Response
Response to Reviewer 3 Comments
Point 1. I recommend this manuscript for publication in IJMS, although before proceeding, I encourage authors to revise some minor issues listed below. 1. Page 1, line 32: Please replace “60 000” to “60,000”. 2. Page 6, line 219: Please replace “nucleotide- dependent” to “nucleotide-dependent”.
Response 1: First of all we would like to thank the Reviewer for the positive review.
As suggested by the Reviewer, in the revised manuscript, we have replaced 60 000 with 60,000 (lane 32) and “nucleotide- dependent” with “nucleotide-dependent” (lane 219).
Reviewer 4 Report
Dear authors, Figure 2 seems slightly grainy; please substitute
In lines 59 and 287 you wrote "since ClpB 59 does not exist in animal cells"; please explain better this point
Author Response
Response to Reviewer 4 Comments
Point 1. Dear authors, Figure 2 seems slightly grainy; please substitute In lines 59 and 287 you wrote "since ClpB 59 does not exist in animal cells"; please explain better this point.
Response 1: We would like to thank the Reviewer for the positive review.
As suggested by the Reviewer, in the revised manuscript, we have replaced Figure 2 with a new one, and we have also added this information (lanes 55-56): It is important to note that Hsp100 members are found in bacteria, protozoa, yeast and plants, but not in animals and humans.
Round 2
Reviewer 1 Report
Authors have made efforts to some extent to cover the gaps indicated in the review/comment. Though the coverage is not up to as a large quantum of published work are well evident from the literature, are acceptable.